# PRAPNet: A Parallel Residual Atrous Pyramid Network for Polyp Segmentation

**DOI:** 10.3390/s22134658

**Published:** 2022-06-21

**Authors:** Jubao Han, Chao Xu, Ziheng An, Kai Qian, Wei Tan, Dou Wang, Qianqian Fang

**Affiliations:** 1School of Integrated Circuits, Anhui University, Hefei 230601, China; p20201029@stu.ahu.edu.cn (J.H.); p20301227@stu.ahu.edu.cn (Z.A.); p20201085@stu.ahu.edu.cn (K.Q.); p20301226@stu.ahu.edu.cn (W.T.); p20301228@stu.ahu.edu.cn (D.W.); p20201087@stu.ahu.edu.cn (Q.F.); 2Anhui Engineering Laboratory of Agro-Ecological Big Data, Hefei 230601, China

**Keywords:** medical image analysis, semantic segmentation, colonoscopy, polyp segmentation, deep learning, health informatics

## Abstract

In a colonoscopy, accurate computer-aided polyp detection and segmentation can help endoscopists to remove abnormal tissue. This reduces the chance of polyps developing into cancer, which is of great importance. In this paper, we propose a neural network (parallel residual atrous pyramid network or PRAPNet) based on a parallel residual atrous pyramid module for the segmentation of intestinal polyp detection. We made full use of the global contextual information of the different regions by the proposed parallel residual atrous pyramid module. The experimental results showed that our proposed global prior module could effectively achieve better segmentation results in the intestinal polyp segmentation task compared with the previously published results. The mean intersection over union and dice coefficient of the model in the Kvasir-SEG dataset were 90.4% and 94.2%, respectively. The experimental results outperformed the scores achieved by the seven classical segmentation network models (U-Net, U-Net++, ResUNet++, praNet, CaraNet, SFFormer-L, TransFuse-L).

## 1. Introduction

Colorectal cancer is widely believed to strike middle-aged adults and is a cancer with one of the highest death rates globally [1,2]. Polyps grown in the gut are one of the early symptoms of this cancer. If not removed in a timely manner, the polyps can cause colon cancer [3,4]. Therefore, detecting and diagnosing intestinal polyps is one of the primary methods of avoiding colorectal cancer [5]. Currently, clinicians mostly depend on an endoscopy to detect and diagnose polyps [6]. However, an endoscopy occasionally neglects cancer-causing polyps, mainly due to the lack of experience of the physicians. As a result, it is critical to build a computer-aided system [7,8] that can precisely detect the positions of polyps in the endoscopic video stream [9,10] and, if the clinician neglects the polyp area, the system can operate another round of scanning, redirecting the clinician to analyze the lesion information in this location.

There are currently only a few public datasets of intestinal polyps available for model training and testing. Polyps exist in various shapes and colors as well as textures [11]. As shown in Figure 1, the minute difference between polyps and the normal ambient environment or partially diseased areas covering the feces is the main challenge in the detection of polyps. In this paper, we used the open datasets of Kvasir-SEG [12] and CVC-ClinicDB and performed relevant experiments

With the expanding utilization and further development of deep learning, a growing number of deep learning-based segmentation methods have recently been proposed [13,14,15,16,17]. Although these methods have made progress, they can only label the detected polyps using bounding boxes and cannot effectively determine the polyp boundary contours, resulting in a poor user experience. Brandao et al. [18] proposed a pre-trained model based on an FCN network to identify and segment polyps to solve this problem. Akbari et al. [19] proposed an advanced FCN-based network to further improve polyp segmentation accuracy.

Since a U-shaped network architecture [14,20,21,22,23] was proposed, its proposed codec structure has been widely used in the field of medical image segmentation. Many newly proposed image segmentation networks continue to follow the core design ideas of the U-network architecture to improve the performance of medical image segmentation tasks by adding new modules or incorporating other design concepts. U-Net++ and ResUNet++ [17] were developed based on the U-Net network structure. Although they achieve ideal results in the polyp segmentation task, these methods focus on segmenting the overall contour of the polyp and tend to ignore the constraint of the polyp region boundaries, resulting in polyp segmentation with rough edges, which is crucial for improving the performance of segmentation. Moreover, the training process of these networks requires a significant amount of time and is difficult to converge.

After Vaswani et al. [24] proposed transformer architecture in 2017, an increasing number of models based on transformer architecture have been proposed for medical image segmentation with proficient results in recent years [25,26]. However, the transformer loses the location information of the original image when converting the polyp image into a word vector for the analysis. The position information is crucial for the polyp segmentation task. Transformer architecture is inferior to a fully convolutional network in the local information acquisition of polyp images. The transformer model is a combination of a few residual modules and layer normalization modules. The most common transformer models today use the layer normalization module, which is located between two residual modules. Therefore, the final output layer has no direct connection path to the previous transformer layer and the gradient flow is blocked by the layer normalization module, which often leads to the problem of disappearing gradients in the top layer during the model training.

Inspired by a fully convolutional network structure [18,19,27,28,29] and a residual network structure [30], in this paper we present a novel deep neural network model for the polyp segmentation task: the parallel residual atrous pyramid network (PRAPNet). We used the residual network as our backbone network for extracting the common features. The ResNet50 network was used due to computing resources and the operation time. Atrous convolution [31] has previously shown its prominence in obtaining global contextual information. To further explore the information contained in each region of the image, we designed a new parallel residual atrous pyramid module (PRAP Module). To further refine the segmentation contour, we applied an attention module [24] and a conditional random field [32]. Table 1 presents the advantages and disadvantages of the model proposed in this paper and the current state-of-the-art models. The main contributions of this paper are as follows.

(1)We offer a novel parallel residual atrous pyramid module for accurate polyp segmentation. Including this module in the FCN-based pixel-level intestinal polyposis prediction framework allows for a more accurate and precise segmentation of the intestinal polyp area, which can greatly enhance the segmentation results when compared with other state-of-the-art approaches. Our proposed improvement can make an effective use of a tiny quantity of picture data.(2)A practical system is established for the intestinal polyp segmentation task that includes all the key implementation details of the method in this paper.

The paper is organized as follows. Section 2 details the methodology proposed in this paper, Section 3 presents the experiments and results, and Section 4 discusses the experimental results. Finally, the conclusions are given in Section 5.

## 2. Methods

### 2.1. Proposed Network Structure

The PRAPNet architecture was based on fully convolutional neural network architecture. A pre-trained residual network, ResNet50, was employed in the network coding section to encode the characteristics of the intestinal polyp images. The input image size was 480 × 480 pixels and the output feature map size was 2048 × 60 × 60 pixels. The feature map was then inputted into the parallel residual atrous pyramid module to obtain another feature map that aggregated different regional context information and further global information. The segmentation results were refined by the attention block and conditional random field. Figure 2 depicts the structure of the network model described in this paper.

### 2.2. Parallel Residual Atrous Pyramid Model

Current mainstream medical image segmentation networks usually rely on U-Net and U-Net-derived networks (e.g., U-Net++ and ResUNet). These models are essentially encoder–decoder frameworks and the usual practice is to concatenate the multi-level features without fully exploiting the contextual information contained in each region of the image. For medical images, the differences between the background area and the area of interest are not very obvious. This problem becomes more prominent in intestinal endoscopy images. The intestinal polyp area and the background area are very similar and are difficult to distinguish. Thus, the intestinal context information of various regions of the polyp images becomes particularly important for the accurate identification of polyps. Considering the above-mentioned situation, improving the ability of a model to identify intestinal polyps is closely related to whether the model can fully utilize the contextual information and the overall body information obtained by the receptive field of the model.

The size of the receptive field of a deep neural network can roughly represent to what extent the researchers can utilize the contextual information. Although the theoretical receptive field of ResNet [30] is already larger than the input picture, previously reported results [34] indicate that the empirical receptive field of a CNN is substantially smaller, especially with higher layers. As a result, many networks are unable to properly exploit the overall context information to improve the model performance. Therefore, we present an efficient overall prior module to address this issue.

The training complexity of a neural network grows as the number of network layers increases. Simultaneously, during the network training phase, the network model will degenerate [15,30]. Previous published research presented a residual architecture to reduce the possibility of network model degradation during training, which enhanced channel dependency whilst lowering the computing cost. This inspired us to propose the model introduced in this paper.

Although low-level features contribute less to performance, they demand more processing resources than high-level features due to their higher spatial resolution. Hence, we created a branch to extract a greater number of high-level characteristics in this module, as shown in Figure 3. A squeeze and excitation unit [33] was also applied to improve the feature encoding to fully exploit the obtained overall information.

A pyramid convolution can obtain the features from each layer of the image and this overall prior design can effectively obtain the context information from each area in the intestinal polyp image. An overall prior branch was designed based on a pyramid convolution in the parallel residual atrous pyramid module proposed in this paper. As shown in Figure 3, this branch used four convolution kernels of different scales to fuse the features. The number of parallel atrous convolutional layers could be adjusted according to the various needs of the experiments. To ensure the weight of the overall features, a 1 × 1 convolutional layer was added after each pyramid layer to reduce the obtained feature dimension to 1/4 of the original feature dimension. All pyramid features were then bilinearly upsampled to restore them to the same size as the original feature map. Finally, the pyramid features of the different scales were fused as the final pyramid pooling feature.

### 2.3. Attention Units

The attention mechanism [24] focuses on a subset of its input and it is widely used in the application of natural language translations. In recent years, it has been used for semantic segmentation tasks [35,36] such as pixel prediction. The attention mechanism helps neural networks to determine which parts of the network need more attention. It also reduces the computational cost of encoding the information in each polyp image into a vector of fixed dimensions. The main advantage of an attention mechanism is its simplicity, which can be applied to any size of the input. The quality of the features can be improved and the results are enhanced. Inspired by this, we implemented attention blocks in the decoder part of the architecture to focus on the basic regions of the feature map.

### 2.4. Conditional Random Field

Conditional random fields (CRFs) are a popular statistical modeling method; this method was applied in this paper to assist with the task of medical image segmentation. CRFs can model useful geometric features such as the shape, regional connectivity, and contextual information to improve the overall results. A CRF was utilized in this work as a further step to produce a more refined output on the prediction map to improve the segmentation results.

## 3. Experiment

### 3.1. Data Preprocessing

For the polyp image segmentation task, every pixel in the training image was labeled as either polyp or non-polyp. The evaluation of the PRAPNet was accomplished utilizing the Kvasir-SEG dataset and CVC-ClinicDB [17] dataset. The Kvasir-SEG dataset consists of 1000 polyp images and their corresponding mask maps annotated by specialist endoscopists from Oslo University Hospital, Norway. The CVC-ClinicDB dataset consists of 612 polyp images. Figure 1 shows example images from the Kvasir-SEG dataset and their corresponding templates. As the number of polyp pictures was too small, direct training would lead to overfitting. Thus, it was necessary to increase the number of polyp pictures in the training set; the number of Kvasir-SEG images was 11,000 and the number of CVC-ClinicDB images was 6732. Traditional data enhancement technology—namely, vertical flip, horizontal flip, 90 degrees clockwise rotation, translation, changing the image brightness, and Gaussian blur—was used to increase the training samples from the initial images. Of these images, 80% were randomly selected for the training, 10% were used for the validation, and 10% were used for the testing.

### 3.2. Implementation Details

The experiments were performed on an Intel(R) Core(TM) i7-7700K with a CPU of 4.20 GHz and a GTX1080Ti graphics card with 11 GB of video memory. PyTorch 1.9.0 and Ubuntu18.0 LTS software environments were used for the neural network experiments. The input of the network was set to 3 × 480 × 480 with a batch size of 4. The training epoch was also set as an iteration of 2200 batches for a total of 100 epochs. In our experiment, training was stopped early after 50 epochs with no significant improvement in validation loss. The network model proposed in this paper was optimized using an ADAM optimizer; the initial learning rate was set to 1 × 10^−5^. Although a lower learning rate slowed down the convergence of the model, it was preferred because a larger learning rate tended to cause convergence failures.

The datasets used in the experiments in this paper contained different image resolutions. To utilize the GPU more effectively and to reduce training time, the uniform size of the image was set to 480 × 480 and then the image was cropped. We utilized 80% of the dataset for the model training, 10% for the validation, and 10% for the testing. We believed that different learning rates and training periods could have an impact on the model training. Thus, to explore the most appropriate parameters for further experiments, comparative experiments with different parameters were conducted.

### 3.3. Experimental Metrics

To verify the effectiveness of our proposed PRAPNet network, the experiments were conducted using the Kvasir-SEG dataset and the CVC-ClinicDB dataset. For a model comparison, the seven most popular segmentation networks (e.g., U-Net, ResUNet, and ResUNet++) were chosen for the comparison. The mean intersection over union (mIoU) and the dice coefficient were selected as the indicators of the quality of the network model.

### 3.4. Experimental Results

Different hyperparameters were applied to optimize the PRAPNet architecture. Hyperparameter tuning was manually performed by training the model with a different set of hyperparameters and evaluating the results. The selection of hyperparameters mostly relied on experience. Too large or too small hyperparameters could be detrimental to the training model. To choose the appropriate hyperparameters, we selected three orders of magnitude for the comparison. Figure 4 is a graph of the loss drop of our model over the iterations of the training cycle with our given hyperparameters. Referring to the existing experience of hyperparameter design, this experiment was conducted to compare three learning rates, lr = 1 × 10^−3^, lr = 1 × 10^−4^, and lr = 1 × 10^−5^. We observed that when the learning rate was set to 1 × 10^−3^, the loss has three peaks, indicating that the learning rate was too large and the gradient oscillated. During the training process, the model was unable to effectively learn the relevant experience because the learning rate was too large; Figure 4 demonstrates that the loss function of the model could not be reduced to a satisfactory level. Figure 4 also demonstrates that when the learning rate was set to 1 × 10^−4^, the loss function of the model showed a rapid decrease at the beginning and the most significant decrease in the first 20 rounds of training. Therefore, we judged that the learning rate of the model could not be set to greater than 1 × 10^−4^. To determine the final learning rate, we set the learning rate to 1 × 10^−5^ when the loss decreased the fastest; the model reached a convergence at an epoch = 30. However, the difference between the value reached by the loss function and the learning rate of 1 × 10^−4^ was very small so we did not consider it necessary to choose a smaller learning rate.

Figure 5 is a comparison of the direct prediction map of the model and the conditional random field after refining the boundary. A conditional random field can effectively fine-tune the boundary of a segmented image. In our work, a high-density conditional random field was applied to the experiment.

Table 2 shows a comparison of the model indicators for three different learning rates. Through the comparison of the indicators, we discovered that the model had the best performance when the learning rate was set to 1 × 10^−5^.

Table 3 and Table 4 present the metric scores obtained by the different models and PRAPNet from the datasets of Kvasir-SEG and CVC-ClinicDB. The comparison network models in this article were all derived from a public code and trained with default parameters. Table 3 shows that, for the Kvasir-SEG dataset, our proposed model outperformed all other three metrics. It should be noted that the dice coefficient reflected the gap between the segmentation results of our model and the actual results. Thus, it was a key parameter and was considered for a further evaluation. In this case, our model achieved a 0.63% improvement over the state-of-the-art SFFormer-L model in terms of the dice coefficients, a 2.4% improvement over the TransFuse-L and CaraNet models, a 4.4% improvement over the PraNet model, and a 12.87% improvement over the classical model, ResUNet++.This proved that our architecture exceeded the baseline architecture. The large class differences and complex backgrounds of the samples in the CVC-ClinicDB dataset and the smaller number of samples in the CVC-ClinicDB dataset compared with the Kvasir-SEG dataset led to the average performance of the model proposed in this paper from the CVC-ClinicDB dataset. This indicated that the proposed model had room for improvement in terms of its class imbalance and model generalization ability. The model generalization problem is common in many models. Subsequent improvements will be made in these two directions.

Figure 6 shows the qualitative results for all models. Figure 7 shows the highlighting of TN, TP, FP, and FN by assigning different colors to the pixels of each category, where TN is black, TP is white, FN is red, and FP is blue. In Table 3 and Figure 6, the advantages of the PRAPNet over the baseline architectures are shown. The quantitative and qualitative results all show that the PRAPNet model trained on the Kvasir-SEG dataset showed satisfying results and transcended the other seven models in terms of the dice coefficient, mIoU, and precision. Therefore, in the task of medical image segmentation, the PRAPNet architecture has obvious advantages over the other segmentation methods.

## 4. Discussion

The PRAPNet architecture proposed in this paper achieved satisfactory results from the Kvasir-SEG dataset. From Figure 6, it could be concluded that, from the Kvasir-SEG dataset, the segmentation maps generated by the PRAPNet outperformed the other architectures in capturing the shape information, demonstrating that the segmentation masks generated in the PRAPNet showed more precise information in the target area than the existing models. The full convolutional network has room for improvement in capturing the polyp location and edge details.

In this paper, we utilized the cross-entropy loss function and the dice loss function to train the proposed model. With the same loss function, the proposed model achieved higher dice coefficient values than the other models. Based on the empirical evaluations, the dice coefficient loss function was chosen to obtain better segmentation results. Additionally, the number of filters, batch size, optimizer, and loss function were observed to affect the results.

We speculated that the performance of the model could be further improved by increasing the size of the dataset, applying more enhancement techniques, and applying a few post-processing steps. We added a small number of parameters to ResNet50, but achieved a higher performance. The application of the PRAPNet should not be limited to biomedical image segmentation; it can also be extended to natural image segmentation and other pixel classification tasks that require further detailed validation. Our work was performed on a NVIDIA-1080ti machine and was limited by the computing power of the machine; the image size was also adjusted. A larger downsampling rate was used for the training, but could lead to a loss of useful information.

Future improvements include:(1)Integrating the parallel residual atrous pyramid module proposed in this paper into other polyp segmentation models to verify its enhancement of the model results.(2)Using a larger batch size and a smaller downsampling rate on a platform with more computing power for optimal training (the experiments in this paper were limited by the computing power of the machine).(3)Optimizing the model by referring to other excellent design concepts to achieve an improvement in performance.

## 5. Conclusions

In this paper, we proposed the PRAPNet, an architecture for the segmentation of intestinal polyps. The proposed architecture utilized a pre-trained ResNet50 model as well as attention units, conditional random fields, and a parallel residual atrous pyramid module. A comprehensive evaluation using publicly available datasets showed that the proposed PRAPNet architecture outperformed the state-of-the-art U-Net and ResUNet++ architectures in generating semantically accurate prediction graphs. To achieve the goal of generalizability, the architecture proposed in this paper should be further investigated for improvements to obtain better segmentation results.

## Figures and Tables

**Figure 1 sensors-22-04658-f001:**
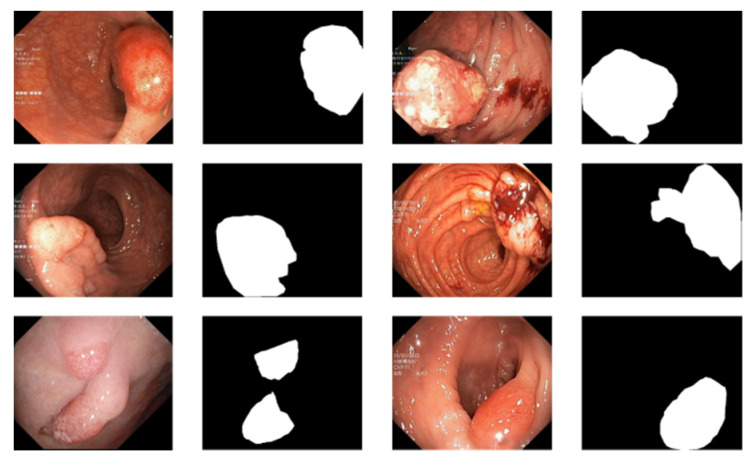
An example of polyp pictures from the Kvasir-SEG dataset, together with their accompanying masks. The first and third columns show the original image; the second and fourth columns show the accompanying masks.

**Figure 2 sensors-22-04658-f002:**
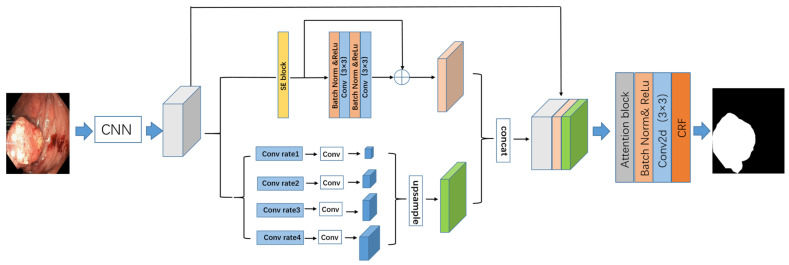
The parallel residual atrous pyramidal network architecture proposed in this paper.

**Figure 3 sensors-22-04658-f003:**
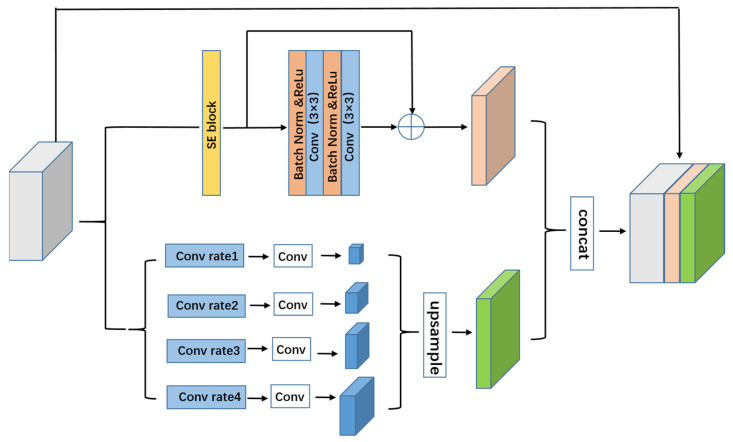
Parallel residual atrous pyramid model.

**Figure 4 sensors-22-04658-f004:**
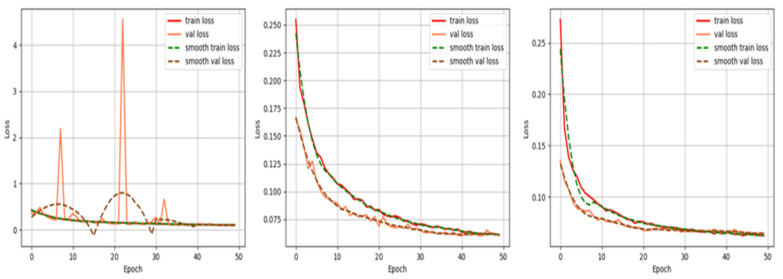
Graph of the loss decline of the PRAPNet model as iterates over the training epochs. The decreasing plots of loss functions for lr = 1 × 10^−3^, lr = 1 × 10^−4^, and lr = 1 × 10^−5^ are shown in order.

**Figure 5 sensors-22-04658-f005:**
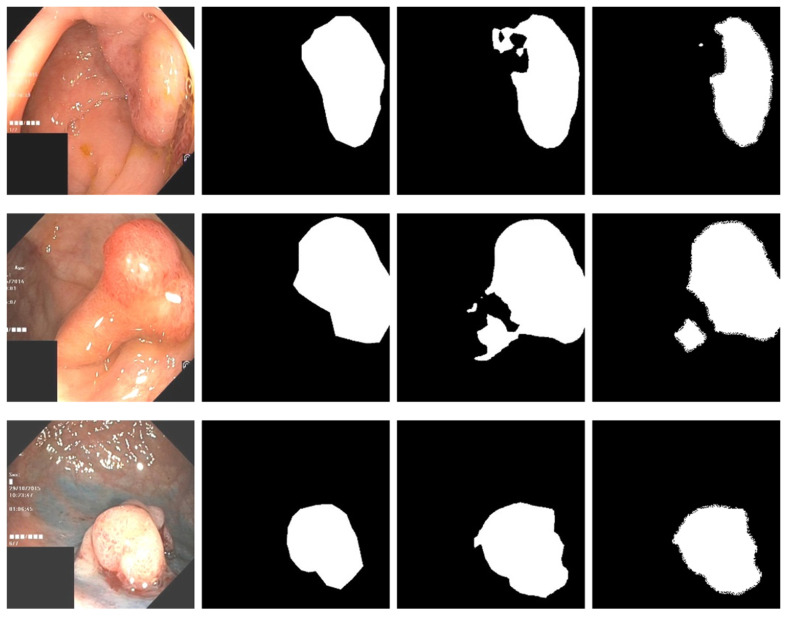
The first column is the original input image, the second column is the mask map, the third column is the direct prediction map of the model, and the fourth column is the prediction map with the conditional random field added.

**Figure 6 sensors-22-04658-f006:**
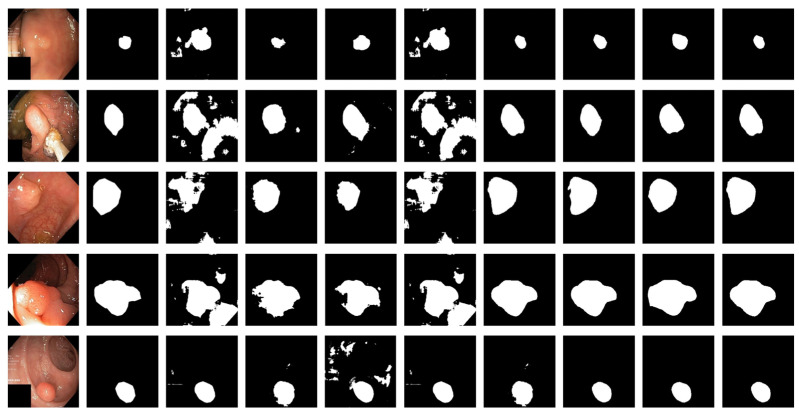
Comparison of qualitative results from the Kvasir-SEG dataset. From left to right are the image, the mask, and the segmentation results of unet, U-Net++, ResUNet++, praNet, SFFormer-L, TransFuse-L, CaraNet, and PRAPNet.

**Figure 7 sensors-22-04658-f007:**
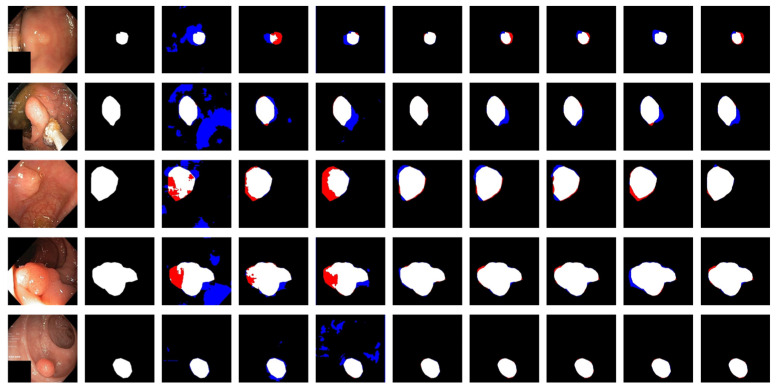
Highlighting of TN, TP, FP, and FN by assigning different colors to the pixels of each category. From left to right are the image, the mask, and the segmentation results of unet, U-Net++, ResUNet++, praNet, SFFormer-L, TransFuse-L, CaraNet, and PRAPNet.

**Table 1 sensors-22-04658-t001:** Strengths and weaknesses of the current state-of-the-art models and the model proposed in this paper.

Method	Strength	Weakness
**SFFormer-L** [24]	These transformer-based models have the perceptual field of the entire image and can take advantage of global contextual information	These models are not as successful at acquiring local information as CNNs and do not take full advantage of the local information on different scales
**TransFuse-L** [25]		
**U-Net** [14]	These models take full advantage of global contextual information and enhance the perception of local contextual information	These network models do not take into account the balance between global and multi-scale information and there is a large number of repetitive operations
**U-Net++** [33]		
**ResUNet++** [17]		
**PraNet** [34]	These two models fully exploit the edge information using the reverse attention mechanism, making the segmentation results appear with fine edges	These two network models mainly focus on the edge information of polyps and do not fully utilize the contextual information of different regions
**CaraNet** [32]		
**PRAPNet**	The model proposed in this paper fully takes into account the global contextual information and multi-scale contextual information of different regions	Due to the use of a two-branch architecture in the proposed decoding module, additional computation may be added by channel redundancy

**Table 2 sensors-22-04658-t002:** Indicator values of models at different learning rates.

Learning Rate	Dice	mIoU	Precision
**1 × 10^−3^**	0.901	0.856	0.902
**1 × 10^−4^**	0.942	0.901	0.934
**1 × 10^−5^**	0.942	0.904	0.936

**Table 3 sensors-22-04658-t003:** Evaluation results of different models using the Kvasir-SEG dataset.

Method	Dice	mIoU
**SFFormer-L** [25]	0.9357	0.8905
**PraNet** [34]	0.898	0.849
**TransFuse-L** [26]	0.918	0.868
**CaraNet** [35]	0.918	0.865
**ResUNet++** [17]	0.8133	0.793
**U-Net++** [37]	0.821	0.722
**U-Net** [14]	0.818	0.742
**PRAPNet**	**0.942**	**0.906**

**Table 4 sensors-22-04658-t004:** Evaluation results of different models using the CVC-ClinicDB datase.

Method	Dice	mIoU
**SFFormer-L** [24]	0.9447	0.8995
**PraNet** [34]	0.899	0.849
**TransFuse-L** [25]	0.934	0.886
**CaraNet** [32]	0.936	0.887
**U-Net++** [17]	0.794	0.729
**U-Net** [14]	0.818	0.746
**PRAPNet**	0.917	0.873

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
