# Peer review of "PRAPNet: A Parallel Residual Atrous Pyramid Network for Polyp Segmentation"

_sensors, 2022, doi:10.3390/s22134658_

Round 1

Reviewer 1 Report

The proposed PRAPNet is a powerful model to detect gut polyps via deep learning. The idea of segmenting polyps from images is promising and the reduced number of images needed to train the model make it a powerful tool for a plethora of tasks. I would like to see it working with other segmentation tasks (not medical).

My only question is about the inference time, which is never cited in the paper. Could you provide an average inference time alongside the other accuracy metrics? This information could be critical in other application fields which require real-time decision making.

Below my english corrections:

line 31: it is critical to build*

line 82: ResNet50

line 119: and can be hardly distinguished*

line 134: remove the bracket near "presented" or add the correct reference missing here

line 134: replace "lower the possibility" with "reduce the possibility"

line 237: In* our work

line 285: ResNet50* 

line 311-318: I think that the author contributions paragraph should only contain text from "Methodology" (line 312) to "manuscript." (line 317)

Reviewer 2 Report

The authors developed a PRAPNet (parallel residual atrous pyramid network) for polyp detection and segmentation in this study. A parallel residual atrous pyramid module to segment the intestinal polyp extracts the global contextual information. Overall, the approach is well defined, and the proposed network improved the segmentation accuracy.  The experimental methodology is well explained, and the results are compared with the existing state-of-the-art approaches. However, some major revisions are required.

1) Please carefully check the text in the figures. upsample and CONCAT is not aligned with the other text in figures 2 and 3. 

2) After the introduction section, please include a comparison table that should include the strengths and weaknesses of the proposed method as well as previous state-of-the-art methods. 

3) Authors claimed the PRAPNet is a lightweight network, please include a table that should compare the number of parameters in your method and the other methods used for similar problems. 

4) In the results section, please include some examples of the good and bag segmentation by highlighting the TN, FP, and FN through different colors assigned to the pixels of each category. 

5) Please include the reference numbers of the related methods in Table 2 and Figure 6. 

6) Is there any special reason to select "Kvasir-SEG dataset" for evaluating your proposed method? Please consider at least one or two more datasets to evaluate your method. 

Reviewer 3 Report

Very well written paper. In the paper, a neural network based on the parallel residual atrous pyramid model for the segmentation of intestinal polyp detection is proposed.   The proposed architecture is based on the fully convolutional neural network architecture. The full use of the global contextual information of different regions by the proposed pyramid module is described. Experimental results show that the proposed global module can effectively achieve better segmentation results in the intestinal polyp segmentation task compared to previously published results. The experimental results outperformed the scores achieved by the classical segmentation network models.

Round 2

Reviewer 2 Report

Thank you for your response. It has raised a few more questions.

1) Your proposed PRAPNet was performing well on the Kvasir-SEG dataset. But it could not maintain the performance on CVC-ClinicDB. Can you explain the reason behind it? And how it can be improved.
2) If your lightweight module i.e. parallel residual atrous pyramid module is not affecting the total number of parameters in the network so there is no point to mention lightweight. It can be confusing to the reader.
3) Mention the reference numbers in all tables and figures as you did in Table 3. It will increase the readability of your manuscript. 
